# Infectious and Commensal Bacteria in Rheumatoid Arthritis—Role in the Outset and Progression of the Disease

**DOI:** 10.3390/ijms25063386

**Published:** 2024-03-16

**Authors:** Aleksandra Korzeniowska, Ewa Bryl

**Affiliations:** Department of Physiopathology, Faculty of Medicine, Medical University of Gdansk, 80-210 Gdańsk, Poland; aleksandra.korzeniowska@gumed.edu.pl

**Keywords:** rheumatoid arthritis, bacteria, pathogens, periodontitis, microbiome

## Abstract

Rheumatoid arthritis (RA) is a chronic, autoimmune disease with a complex outset. Besides the genetic susceptibility in its pathogenesis, various environmental factors also participate. Of these, in recent years, there have been increasing reports of the involvement of bacteria in the disease’s outset and development, especially gut microbiota and oral pathogens. Most recent reports about bacteria participation in RA pathogenesis focus on *Prevotella copri* and *Porphyromonas gingivalis*. There are also reports about the involvement of respiratory and urinary tract pathogens. The exact mechanisms leading to RA development used by bacteria are not well known; however, some mechanisms by which bacteria can interact with the immune system are known and can potentially lead to RA development. The aim of this study is to provide a comprehensive review of the potential bacteria participating in RA development and the mechanism involved in that process.

## 1. Introduction

Rheumatoid arthritis (RA) belongs to a group of chronic autoimmune diseases connecting abnormalities from both adaptive and innate immune systems. This rheumatic disease involves progressive inflammation initially involving small joints before gradually moving to larger and larger joints. The result is the destruction of joint structure, and the inflammation of the synovial membrane by several immune cells, including macrophages, T and B cells, as well as the cytokines and matrix metalloproteinases (MMPs) released by them [1]. The progression of the illness can cause the systemic involvement of cardiovascular [2,3], respiratory [4], and renal tissue [5]. A basic determinant in RA diagnosis is the presence of rheumatoid factors (RFs) and anti-citrullinated peptide antibodies (ACPAs).

A major risk factor for RA outset is genetic predisposition, among which the greatest risk is HLA-DRB alleles from major histocompatibility complex (MHC) molecules. Even though genetic factors definitely increase the risk of disease outset, it does not determine the illness. There are many environmental factors that, in coexistence with genetic factors, can lead to RA development. These factors including age, gender, smoking, and infections [6]. Viral infections were suspected for their importance in rheumatic disease development for years. There are various reports about the Epstein–Barr virus (EBV) [7,8], Cytomegalovirus (CMV) [9], Parvovirus B19 [10], or even Human Herpes Virus 6 (HHV-6) [11] and Human Immunodeficiency Virus (HIV) [12] occurring in higher frequencies in patients with rheumatic diseases.

Bacterial infections are a relatively new topic in research about the exact pathogenesis of rheumatic diseases, including RA. Recently, researchers have provided a lot of evidence for a potential link between bacteria and RA outset. It is known that some pathogens can influence immune response by their virulence mechanisms. Notably, bacteria are a rich source of different antigens which can be recognized by Toll-like receptors (TLRs) and lead to various autoimmune processes [13]. Lipopolysaccharide (LPS) produced by bacteria due to biofilm formation can also stimulate innate immune receptors in synovium, which leads to joint inflammatory response and degradation [14]. Understanding the role of bacteria in RA development and the discovery of the exact mechanisms involved in RA pathogenesis could improve actual treatment, diagnosis, and prevention methods.

The aim of this study is to provide a comprehensive review of the recent knowledge about the possible participation of different infectious and commensal bacteria in RA pathogenesis and development. The data presented in this review were sourced from the most recent studies about this topic (up to May 2023). PubMed, Scopus, Science Direct, and Google Scholar were searched using the following expressions: rheumatoid arthritis, bacterial infections, periodontitis, and microbiome.

## 2. Immunopathogenesis of Rheumatoid Arthritis

Although the mechanism of pathogenesis remains unclear, there are a lot of factors potentially involved in the disease’s outset, and genetic factors are one of the greatest risk of rheumatoid arthritis development. Even though the presence of genetic factors is essential for RA development, it does not prejudge it. Between relatives, the probability of disease occurrence is around 40–50%, with the risk increasing if a first-degree relative has the condition, while between unrelated people, it is around 1%. Also, ethnicity has a key impact on RA development [6]. The major genetic influence is assigned to class II histocompatibility complex molecules. The connection between Human Leukocyte Antigen (HLA) and the occurrence of RA was observed decades ago. Many different diseases have been reported to correlate with HLA. The reason for this connection is probably because of the high heterogeneity of this region [15]. Most of the autoimmune diseases associated with HLA correlate with HLA class I and HLA class II alleles. In the case of RA, HLA class II molecules are the main genetic factors [16]. HLA-DRB1 alleles (*0401, *0402, *0405, and *0101, *102) are known to strongly correlate with the probability of the outset and exacerbation of RA. Most of these alleles have a common conserved amino acid sequence which is located at positions 70–74 in the third hypervariable region of the DRβ1 chain. In the shared epitope (SE) hypothesis, HLA-DRB1 alleles have a direct impact on RA susceptibility and disease progression [17].

There is a strong connection between SE-coding DRB1 alleles and anti-cyclic citrullinated peptides (anti-CCPs), which are specific markers of RA. ACPAs can be produced years before the disease outset; the presence of ACPAs in undifferentiated arthritis indicates a possible progression toward RA. RA patients, in comparison to undifferentiated arthritis patients, display a wider isotype profile (IgG, IgM, IgA, and IgE ACPA). ACPA-positive RA differs significantly from ACPA-negative RA in severity and prognosis [18]. The association of SE-coding DRB1 molecules with RA only concerns ACPA-positive RA patients. This indicates that this molecule participates in RA development and progression by enabling ACPA production. The connection between RA, ACPAs, and HLA-DR has not been explained yet. At this point, it has not been demonstrated whether fibrinogen peptides bind preferably to SE-coding HLA-DR molecules [19].

The production of ACPAs occurs as a result of tolerance breaks. Citrullination is a physiological process of post-translational modification during which arginine residues of proteins (vimentin, fibrinogen, and histone [20]) are converted to citrulline by peptidylarginine deiminase (PAD). Among the five PAD isoforms, only two (PAD2 and PAD4) are involved in RA. Citrullination can create specific neoantigens that activate T lymphocytes, which facilitate the activation of B cells and their differentiation into plasma cells, resulting in ACPA production. Apart from SE coding, HLA-DR molecule citrullination also increases due to other factors, including smoking and infections [21].

Many immune cells are implicated in RA pathogenesis [22] (Figure 1). In RA, there is an overproduction of specific pro-inflammatory cytokines and a diminished production of anti-inflammatory cytokines. Some of these cytokines, like interleukin-17 (IL-17), tumor necrosis factor (TNF-α), and interleukin-1 (IL-1), contribute to RA pathogenesis. Th17 cells, a subset of T cells, secrete various pro-inflammatory cytokines, including IL-17, TNF-α, granulocyte-macrophage colony-stimulating factor (GM-CSF), interleukin-21 (IL-21), and interleukin-22 (IL-22), and are seemingly key mediators in RA pathogenesis [23,24]. Th17 cells, in normal conditions, produce IL-17 and IL-22 cytokines as a from of protection against extracellular bacteria, fungi, and mycobacteria and remains in balance with Treg cells. When this balance is disturbed, it can lead to the development of autoimmune diseases [25]. The level of Th17 cells was elevated in peripheral blood mononuclear cells (PBMCs) isolated from early-RA patients in comparison to healthy controls in [26]. The pro-inflammatory cytokines produced by Th17 cells activate osteoclasts, neutrophils, and monocytes and recruit lymphocytes. In effect, leukocytes produce chemokines, cytokines, and enzymes, resulting in chronic inflammation, bone destruction, and cartilage damage in RA [27]. In the context of bacteria participation among all cell types, the key role at the outset of the disease seems to be played by neutrophils. Their pathogenic role is the sum of various processes, including the generation of pro-inflammatory cytokines, chemokine receptors, and anti-apoptotic molecules; they also enhance oxidative stress and release neutrophil extracellular traps (NETs) [28].

## 3. The Connection between Periodontal Disease and Rheumatoid Arthritis

Periodontitis (PD) is an inflammatory disease arising as a result of the host immune response to biofilm-forming microorganisms. The progression of PD leads to the destruction of the periodontium and may result in tooth loss [29]. The destruction of the periodontium occurs because of the ‘pocketing’ process, which involves the formation of a pocket between the gingiva and teeth [30]. This is a disease with a complex pathogenesis, in which pathogens, commensals, and host oral autoimmunity dysfunction are involved [31]. PD was frequently proposed as a risk factor in various systemic disorders, like diabetes [30,32], cardiovascular [33] and respiratory diseases [34], adverse pregnancy outcomes [35], and rheumatic diseases, including RA. Research on this topic indicates that there is a higher prevalence of PD among RA patients [36,37]. The relation between RA and PD could be due to their similar environmental (e.g., smoking) and genetic risk factors. The similarities between PD and RA are apparent on many levels (Table 1), so the co-occurrence of these diseases can affect the deterioration of patients’ conditions. Another possibility is that RA predisposes to PD. On the other hand, PD can also be an RA risk factor due to various pathways, including the activity of periodontal bacteria [38]. The bacteria involved in the initiation of PD due to the red complex theory are *Poprhyromonas gingivalis* (*Pg*), *Tannerella forsythia* (*Tf*), and *Treponema denticola* (*Td*) [39]. In addition to red complex bacteria, *Filifactor alocis* (*Fa*), *Synergistetes*, *Peptostreptococcaceae*, and *Aggregatibacter actinomycetemcomitans* (*Aa*) also participate in PD initiation [31]. *Poprhyromonas gingivalis* (and *Aggregatibacter actinomycetemcomitans* are major PD pathogens potentially involved in RA outset and development.

There are many reports about the occurrence of specific periodontal bacteria among RA patients, among which *Porphyromonas gingivalis* is mentioned most frequently [51,52,53]. Some scientists suspect that *Pg* citrullination activity can lead to autoimmunity in RA. The citrullination process occurs due to the enzyme peptidylarginine deiminase (PAD), which transforms L-*arginine* into citrulline. This bacterial enzyme is capable of facilitating the citrullination of both self-proteins and host proteins. In RA, the citrullination of autoantigens results in autoimmune response and the production of specific antibodies called anti-citrullinated protein antibodies (ACPAs) [54]. The activity of PAD is elaborated as a major component of *Pg* participation in the initiation of RA. *Pg* is the only known bacteria-produced PAD capable of carrying out the C-terminal citrullination of fibrinogen and enolase peptides. This capability makes the role of this bacteria an interesting subject regarding RA pathogenesis studies. Jenning et al. discovered that beyond carrying out the C-terminal citrullination of *Pg*, PAD is also capable of carrying out the internal citrullination of arginine in fibrinogen/vimentin. The ACPAs produced in RA are recognized in many different citrullinated autoantigens, including these citrullinated by PAD (e.g., 71vim). Recombinant PAD from *Pg* was able to citrullinate major RA autoantigens. Moreover, they confirmed the correlation between the levels of anti-RA–PADs and ACPAs. These discoveries support the possible role of bacteria like *Pg* in the pathogenesis and progression of RA by inducing ACPA production [55]. Maresz et al. observed that the infection of a collagen-induced arthritis (CIA) mouse model with *Pg* led to an earlier outset, a severe course, and an enhanced progression of the disease in comparison to the non-infected animals [56].

Besides ACPAs, there is the linkage between *Pg* and RF, which is also commonly detected among periodontitis patients [57]. RFs are antibodies commonly found in blood samples from patients with RA. They can occur in different forms, among which the most leading are IgM and IgA RFs. RFs recognize the constant part (Fc) of IgG class molecules and bind to them. According to Maibom-Thomsen et al.’s research, RFs do not react with native IgG in solution, and they are circulated nearby them in the blood [58]. Previous studies indicate that the Fc region of IgG can become a perfect target for RFs because of *Pg* proteinase activity, which can specifically decompose arginine and lysine in IgG3 CH2 and CH3 domains [59]. As mentioned above, the cysteine proteinases (known as gingipains) involved in that process are one of the crucial *Pg* virulence factors. Gingipains can be specific to arginine (RgpA or RgpB) or lysine (Kgp). According to this characteristic, they are divided into two types. They are crucial during infection because of their role in host amino acid uptake, fimbriae maturation, hemagglutination, and the degradation of host proteins [60,61]. The results about the association of gingipains and RA are conflicting. In their studies, Svärd et al. discovered that the levels of RgpB antibodies in RA patients were significantly higher than in healthy controls, although they did not find a correlation between these antibodies and ACPA positivity levels [62]. In contrast, Bae et al. not only observed higher RgpB antibody levels in RA patients but also noted a positive correlation between these antibodies and ACPA levels [63]. Some studies did not find any association between these antibodies and RA [64].

Another possible way that *Pg* participates in RA pathogenesis is through TLRs. TLRs are a part of the innate immune system. They act against microorganisms by recognizing pathogen-associated molecular patterns (PAMPs). The activation of TLRs by PAMPs results in a variety of defense mechanisms, like phagocytosis and increased production of inflammatory cytokines, chemokines, reactive nitrogen, and oxygen. TLRs are also responsible for the production of costimulatory molecules, which makes them an important mediator between the innate and adaptive system [13]. In humans, 10 paralogous TLRs are recognized. Each of them is recognized and activated by a specific small group of microbe-delivered molecules [65]. For example, TLR-2, TLR-1, and TLR-6 recognize lipopeptides in bacteria, mycobacteria, and mycoplasma. Also, the effects of PAMP recognition can differ: after activation, TLR-1, TLR-2, TLR-4, and TLR-5 produce specific inflammatory cytokines; on the other hand, TLR-3 and TLR-7 induce the production of type I interferons [66]. TLRs also are responsible for the imitation of various transcription factor activations, including nuclear factor κB (NFκB). In PD, PAMP recognition leads to the overexpression of the receptor activator of the nuclear factor-κB ligand (RANKL) in osteoclasts. In effect, this action, together with cytokine overproduction, leads to osteoclastogenesis [50]. The same situation occurs in RA [48]. A higher expression of TLRs, especially TLR-2, TLR-3, TLR-4, and TLR-7, in the synovial fluid and tissue of early-RA and RA patients was observed in various studies [67,68,69]. It was noticed that LPS from *Pg* activates TLR-2. One of the observed effects of this activation is the upregulation of thrombospondin-1 (TSP1) in monocytes. The higher expression of TSP-1 was also strengthened by inflammatory cytokines such as IL-17. Because of the dual mechanisms of TSP-1, such as its interaction with both CD36 receptors and integrins, it appears that TSP-1 has both pro- and anti-inflammatory functions. Also, because of the anti-angiogenic functions of TSP-1, it can be involved in chronic inflammation outset [70]. Another observed result of LPS recognition by TLR-2 is an enhanced production of interleukin-33 (IL-33). IL-33 upregulation is commonly observed in chronic inflammatory diseases, including RA [71].

The second most common oral bacteria in the context of RA pathogenesis is *Aggregatibacter actinomycetemcomitans* (*Aa*). This Gram-negative red complex bacteria is reported to be the bacteria most commonly isolated from periodontal lesions. The estimated percent of patients with severe periodontitis infected by *Aa* is around 89–100% [72]. One of the basic virulence mechanisms of *Aa* is the production of leukotoxin A (LtxA). *Aa* production can occur in two phenotypes, namely minimally leukotoxic (652 strains) and highly leukotoxic (JP2 strains) strains, the latter of which produce more LtxA and ltx mRNA, although the direct mechanism responsible for the higher production of LtxA by JP2 is not known [73]. LtxA is a member of the repeats-in-toxins (RTX) family, which includes α-hemolysin from *Escherichia coli* and CyaA from *Bordetella pertussis*. Like all family members, for its activity, LtxA requires calcium ions and generates unregulated calcium influx into the cells. This mechanism can be the basis of LtxA cell lysis due to its promotion of calcium changes in T cells, which result in various events, like calpain activation and the mobilization of β2 integrins into membrane lipid rafts [74]. It is believed that LtxA is responsible for damaging leukocyte membranes. The process starts with surface absorption and interactions with function-associated antigen-1 (LFA-1) and other β_2_ integrins on white blood cells, which, by triggering intercellular pathways, leads to cell death. A member of the HACEK group of bacteria, *Aa* could be one of the reasons for the development of various systemic diseases, for example, endocarditis [73]. According to Konig et al., LtxA could be involved in *Aa’s* capability to induce global citrullination in host neutrophiles by mimicking pathways (mediated membranolysis) responsible for autoantigen citrullination in RA. The citrullinome created by LtxA action matched the RA patient’s citrullinome in 44/86 proteins. This observation suggests that *Aa* periodontal infection can be enough for the production of antigenic determinants recognized by autoantibodies specific in RA [75]. Yoshida et al., during their research, discovered the relationship between RA and the heat shock protein from *Aa* (DnaJ). According to their study, patients with RA have significantly higher titers of IgG antibodies against the N-terminal conservative region (J-domain and G/F region) of the DnaJ protein. This indicates that this region can play an etiological role in RA pathogenesis [76].

The discussed correlation between periodontal diseases and RA indicates that the restoration of the periodontium can be a helpful strategy in reducing RA severity. Despite the potential pathogenetic link between PD and RA, patients with RA and PD deal with the same imbalance of cytokines, including higher levels of gingival crevicular fluid of pro-inflammatory cytokines (IL-1β, IL-6, IL-4, and TNF-α) and lower levels of anti-inflammatory cytokines (e.g., IL10) [48,77,78,79].

Non-surgical periodontal treatment is one of the most common treatments used in periodontal therapy. Assumptions of the treatment consist of patient oral hygiene improvement and the mechanical removal of plaque and calculus deposits. This form of treatment aims to limit the population of “red complex” bacteria, and subsequent maintenance should prevent the formation of a population with a size sufficient for disease development to be repeated [80]. Białowąs K. et al. studied the impact of non-surgical periodontal treatment on patients with RA and spondyloarthritis (SpA). The treatment applied resulted in a reduction in disease activity, measured by DAS28, in the RA patients, while the clinical and biochemical parameters did not change among the SpA patients. They observed an improvement in the visual analogue scale (VAS) score and the number of swollen and tender joints in the RA group, which probably influenced the decrease in the DAS28 score [51]. D’Aiuto F. et al., in their study, observed that periodontal therapy significantly decreases the levels of IL-6 and CRP in serum in patients with periodontitis. This is promising due to the cytokine imbalance similarity between PD and RA patients [81]. In a meta-analysis conducted by Sun J. et al., they confirmed that periodontal therapy induced a significant reduction in DAS28, VAS score, CRP, and number of tender and swollen joints. All of these results suggest that the usage of periodontal treatment can decrease the severity of RA and improve patients’ lives [82].

## 4. Respiratory Infections

The common respiratory pathogens belong to *Streptococcus pneumoniae* and *Haemophilus influenzae*. They are commonly found in the nasopharynx of healthy people, where the overgrowth of these pathogens is balanced due to the presence of other commensal bacteria, the majority of which are members of the genera *Corynebacterium*, *Dolosigranulum*, and *Staphylococcus*. The disturbance in this protection mechanism leads to respiratory tract infections [83].

Findings stating that airway abnormalities are associated with RA-related autoantibody positivity in patients without inflammatory arthritis indicate that the lung may be a potential initiating site of RA with similar autoimmunity [84]. Moreover, the fact that smoking significantly increases the risk of RA development supports the hypothesis that the lungs have a potential role in RA pathogenesis [85]. In their research, Villis Van C. et al. detected RA-related antibodies in the sputum of patients with early RA and a group predisposed to RA This result suggests that the lungs may be a place of autoantibody production in early RA [86].

Another potential role for the lungs in RA pathogenesis could be due to the lung microbiome. Using 16S sequencing, Scher U. J. et al. discovered that the lung microbiota of RA patients significantly differ from those of healthy controls, being more similar to sarcoidosis microbiota. In both cases, a decrease in or absence of *Actinomycetaceae*, *Spirochaetaceae*, and *Burkholderiaceae* and the genera *Actynomyces*, *Treponema*, and *Porphyromonas* in comparison to healthy controls was observed. These findings indicate that mucosal inflammation could be one of the factors of lung dysbiosis in both cases. Because this dysbiosis correlates with systemic and local autoimmune changes, it can be implicated in RA pathogenesis in some cases [87].

## 5. The Impact of Intestinal Tract Microbiota Dysbiosis on Rheumatoid Arthritis

The relationship between the intestinal microbiota and autoimmune diseases has been examined often in recent years. The potential impact of microbiota was reported in Hashimoto’s thyroiditis, type I diabetes, multiple sclerosis, psoriatic arthritis, systemic lupus erythematosus, and rheumatoid arthritis [88]. The human microbiome established after birth and during human life increases the diversity of its composition. In adults, the microbiome is composed of several hundred species, with Bacteroidetes and Firmicutes being dominant. Microbiota are responsible for a variety of processes, including metabolic, physiological, nutritional, and immunological processes [89].

Microbial antigens are in continuous communication with the immune system. This communication allows microbes to regulate immune responses through the influence of B cells, innate-like T cells, T helper cells, and T regulatory cell responses. In the case of microbiota dysbiosis, the basis of this interaction is disturbed, which may lead to abnormalities in the functions of the immune system [90]. For example, dysbiosis in the microbiota caused by factors such as bacterial infections, antibiotic therapy, and a poor diet (Figure 2) can lead to abnormalities in the action of innate immune cells which result in the upregulation of pro-inflammatory cytokines (type I interferon (IFN), interleukin-12 (IL-12) and interleukin-23 (IL-23)) and the downregulation of anti-inflammatory cytokines (IL-10 and transforming growth factor β (TGF-β)) [91]. The same cytokine imbalance can be found in the serum, synovial fluid, and synovial tissue of RA patients [92]. The impact of gut microbiota in the pathogenesis of RA is probably multifactorial. The proposed mechanisms include the activation of antigen-presenting cells by TLRs or nucleotide oligomerization domain (NOD)-like receptors (NLRs), the production of citrullinated peptides through enzyme activity, molecular mimicry, and, as mentioned above, the control of immune cell responses [46].

The commensal bacteria residing in the intestines are a rich source of various antigens recognized by TLRs. One of the crucial antigens involved in gut dysbiosis is flagellin, which is a component of the flagellum mostly occurring in Gram-negative bacteria recognized by TLR-5. The activation of TLR-5 by flagellin results in various mechanisms, including the production of pro-inflammatory cytokines, nitric oxide, and chemokines [93]. Moreover, flagellin, through the activation of TLR-5 and the enhanced production of specific cytokines (IL-17, IL-21, and IL-22), promotes Th17 cell production, which results in the disbalance of Th1/Treg cells [94]. Different gut resident bacteria can trigger autoimmune diseases through Th17 cells, and this has been confirmed by various animal model-based studies. Wu Hsin-Jung et al. discovered that the severity of autoimmune arthritis was strongly attenuated in a germ-free mouse model. The germ-free conditions also contributed to a decrease in the Th17 cell population. The monocolonization with segmented filamentous bacteria (SFB) resulted in an enhanced production of Th17 cells, autoantigen production, and the remission of arthritis. This indicates that through the promotion of a specific subset of Th cells, single commensal bacteria can trigger autoimmune disease outset [95]. On the other hand, polysaccharide A (PSA) from *Bacteroides fragilis*, through the activation of TLR-2, enhanced the activation of Treg cells [96].

One of the major gut commensal bacteria in the context of RA is *Prevotella copri*. A high abundance of *Prevotella copri* in the gut microbiota of early-RA patients has been commonly detected [97,98]. The levels of antibodies against *P. copri* correlate to immune response (the production of specific cytokines and chemokines) by Th1 and Th17 cells [99]. In their research, Maeda Yuichi et al. observed that the colonization of a mice model with *P. copri* triggers the development of joint inflammation. Moreover, changing the conditions to germ-free conditions immediately stopped the arthritis development. The proposed mechanism of pathogenesis is the activation of T cells with genetic predisposition by *Prevotella* enrichment in microbiota, which leads to arthritis in mice. The microbiota *P. copri* enrichment also increased the levels of Th17 cells in the large intestine, which could also indicate the role of this bacterial strain in autoimmunity development [100].

Because of all the evidence about the microbiome’s role in RA development and severity, immunosuppressive drugs that can restore microbial gut composition have become potential therapeutic targets in autoimmune rheumatic diseases. Some of the possible beneficial agents in RA treatment are probiotics. Probiotic bacteria can be involved in modulating the immune system by affecting specific cells like dendritic cells (DCs), macrophages, natural killer cells (NKs), and lymphocytes. Moreover, probiotics are capable of downregulating TLR expression, which results in inflammation reduction [101]. There is evidence of the beneficial role of probiotics in animal arthritis models. Treating adjuvant-induced arthritis rats with *Lactobacillus casei* significantly inhibited arthritis development while protecting against bone destruction in [102]. In another study, *Lactobacillus acidophilus* decreased arthritis score and also protected tissues from oxidative stress in a rat arthritis model [103]. Moreover, probiotic treatment is also effective among patients with RA. In their research, Mandel David R et al. observed a reduction in CRP levels, improvement in patients’ self-assessed disability, and an improvement in global assessment scores among RA patients treated with *Bacillus coagulans* [104].

## 6. Urinary Tract Diseases

Urinary tract infections (UTIs) are common, recurrent diseases caused by Gram-negative bacteria, mostly uropathogenic *Escherichia coli* (UPEC). The risk of developing a UTI is influenced by female gender and sexual activity, although the greatest risk factor is UTI history. Besides UPEC bacteria, which are the reason for approximately 80% of UTIs, some Gram-positive bacteria (*Enterococcus faecium* and *Staphylococcus aureus*) and different Gram-negative bacteria (*Proteus mirabilis*, *Klebsiella pneumoniae*, *Enterobacter species*, and *Pseudomonas aeruginosa*) are associated with the typical etiology of UTIs [105].

Regarding the pathogenesis of RA among all the bacteria causing UTIs, the most important are *Proteus mirabilis* (*Pm*). In the 1980s, significantly elevated levels of antibodies against this bacteria were found in blood samples of RA patients in comparison to healthy controls and different autoimmune rheumatic diseases (ankylosing spondylitis, systemic lupus erythematosus) [106,107]. The proposed mechanism by which *Pm* could be involved in RA pathogenesis is molecular mimicry. The sequence EQRRAA (glutamic acid–glutamine–arginine–arginine–alanine–alanine), localized in position 69–74 of HLA-DRB1, is a sequence involved in the shaping of T cell repertoires [108]. There is a molecular similarity between this sequence and the ESRRAL sequence of hemolysin. During their study, Pm. Tiwana et al. observed that an antiserum against the EQRRAA sequence raised in rabbits bound to similar peptides containing the ESRRAL sequence. Additionally, antiserum EQRRAA and ESRRAL bound to a mice fibroblast cell line which expresses HLA-DRB1p0401, which is an allele connected to RA. On the other hand, the aforementioned antiserums did not bind to fibroblasts expressing HLA-DRB1 alleles not associated with RA [109]. A few years later, Rashid T. et al. discovered that the elevated levels of IgG and IgM antibodies to *Pm* and against EQRRAA and ESRRAL peptides, detecting them in both Japanese and Finnish patients with early and advanced RA in comparison to their corresponding healthy controls. This observation confirms the potential participation of *Pm* in RA pathogenesis by molecular mimicry [110]. These results were expanded on by Newkirk M.M. et al., who noticed that the levels of IgM and IgG antibodies to *Pm* were significantly higher in patients with RF-positive RA in comparison to patients with RF-negative RA, spondyloarthropathy, and undifferentiated arthritis. Their results indicate that elevated levels of IgM and IgG antibodies to *Pm* are associated with early seropositive RA and occur only in the presence of antibodies specific for immunoglobulin IgG damaged with advanced glycation end-products (anti-IgG-AGE antibodies) [111]. In a different study conducted by Chandrashekara S. et al., the researchers did not find a significant increase in antibodies to *Pm* in patients with RA compared to healthy controls [112]. The results of this study show that there is no consensus regarding the involvement of Pm in the pathogenesis of RA, and further research is required to determine the role of this bacterium.

On the other hand, the situation can be reversed, and RA can directly influence the occurrence of the UTIs and de novo lower urinary tract symptoms (LUTS), which commonly affect women. Overactive bladder syndrome and urinary incontinence are the most common LUTS among RA patients. One of the risk factors of LUTS occurrence in female patients with RA is a higher BMI [113]. A higher incidence of hospitalization due to UTIs was observed in RA patients, and the major reason for this situation could be the increased frequency of all infections in RA patients. The same as with LUTS, the at-risk group mainly consists of women. Puntis D. et al. discovered that the risk of UTIs grows with taking long-term oral steroids and increasing their dose, while taking methotrexate does not influence the occurrence of infections in RA patients. They also confirmed that a higher incidence of UTIs correlates with RA occurrence and could be due to the complication of RA itself [114].

## 7. Conclusions

The pathogenesis of RA is without question a complex process involving the coexistence of many various factors. Although researchers are not in agreement about which bacteria are implicated in this process and how they are involved, the evidence about the importance of their role seems to be extensive. Infectious and commensal bacteria can trigger immune response through various virulence mechanisms which, in susceptible individuals, can lead to autoimmune response. In the majority of cases, the use of therapy targeting bacterial infections resulted in an improvement of the condition of RA patients. It is worth remembering that the role of bacteria in RA development is one of the factors that can lead to the development of RA, and it is also worth remembering that RA, like many autoimmune diseases, is a multifactor condition. However, the topic of bacterial involvement is still worthy of study, especially regarding research on improving detection methods. The recognition of the microbes implicated in RA and their mechanisms of interaction will be indispensable for future discoveries regarding RA treatment, diagnosis, and prevention.

## Figures and Tables

**Figure 1 ijms-25-03386-f001:**
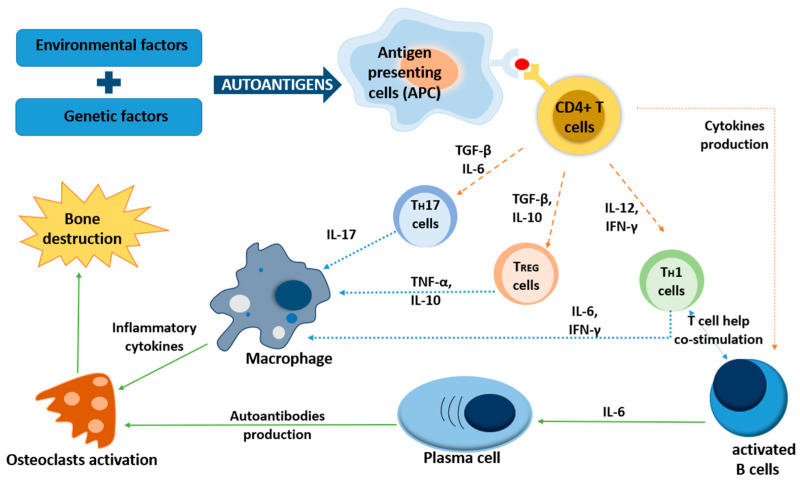
Immunopathogenesis of rheumatoid arthritis. Rheumatoid arthritis outset occurs due to the connection between genetic susceptibility and exposure to different environmental factors (including bacteria). Autoantigens arising from citrullination are presented by antigen-presenting cells (APCs), which are recognized by T cells. After recognition, T cells differentiate into Th1 and Th17 cells, which stimulate macrophages to facilitate inflammatory cytokine production. Cytokines, through the activation of osteoclasts and chondrocytes, lead to joint damage and bone destruction.

**Figure 2 ijms-25-03386-f002:**
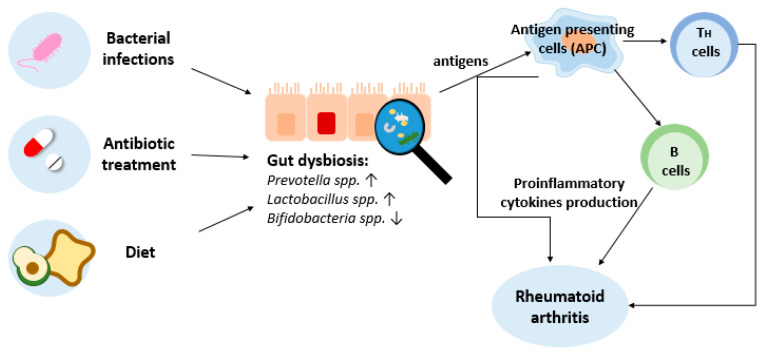
Impact of gut dysbiosis on immune cells, leading to rheumatoid arthritis. ↑ - increase, ↓ - reduction.

**Table 1 ijms-25-03386-t001:** Similarities between periodontitis and rheumatoid arthritis.

Features	Periodontitis (PD) and Rheumatoid Arthritis (RA)	References
Course of the disease	Both are chronic, inflammatory diseases proceeding with the accumulation of immune cells (lymphocytes (B and T cells), monocytes, and neutrophils).	[40]
Genetic factors	Genetic factors play an important role in disease outset in both diseases. Common genetic factors implicated in RA and PD include HLA-DRB3 and HLA-DR4.	[19,41,42,43]
Environmental risk factors	Air pollution, smoking, and gut microbiome dysbiosis are risk factors in both diseases.	[44,45,46,47]
ACPA production	ACPAs are produced in PD due to *P*. *gingivalis* activity. In RA, the production of ACPAs leads to joint inflammation and destruction.	[48]
Cytokines	In both diseases, the upregulation of cytokines and MMPs is involved in pathogenesis. The cytokine profiles in both diseases are similar (upregulation of TNF-α, Il-1β, and IL-6 and downregulation of TGF-β and IL-10).	[40,48,49]
Receptor activator of nuclear factor-κB ligand	The overproduction of the receptor activator of nuclear factor-κB ligand (RANKL) in osteoclasts leads to osteoclastogenesis and bone resorption in both diseases.	[48,50]

## Data Availability

Data are contained within the article.

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
