# Peer review of "Infectious and Commensal Bacteria in Rheumatoid Arthritis—Role in the Outset and Progression of the Disease"

_ijms, 2024, doi:10.3390/ijms25063386_

Round 1
Reviewer 1 Report
Comments and Suggestions for Authors
The present manuscript reviews immunopathogenesis of Rheumatoid arthritis (RA) and relationship between RA and bacterial infection.
The manuscript is very well written, and the topic is very interesting and bacteria participation in RA pathogenesis is clear and well-focused.
However, some major issues need to be coped with to allow publication.
Here are some specific points that need to be addressed:
1) The authors mentioned about red complex involved periodontitis. They state that “red complex theory are mainly Porphyromonas gingivalis (Pg), Filifactor alocis (Fa), Synergistetes, Peptostreptococcaceae and Aggregatibacter actinomycetemcomitans (Aa)”. However, I think red-complex bacteria in periodontitis are Porphyromonas gingivalis (Pg), Tannerella forsythia (Tf), and Treponema denticola (Td). Could you explain the reason why you chose other bacteria for red complex in this manuscript?
2) Table 1 – What are genetic factors related in both disease? The authors should mention about some representative genes related to both PD and RA.
3) Introduction – “Progression of the illness can cause systemic disorders in the heart, blood vessels, kidney, liver, and lungs.”. Could you add the reference for each disease?
Minor comments
1) Page 3, line3 – Please add new reference.
2) Page 5, line21 – Please remove “[“ before Ref. 51.
3) Page 6, line 17 – Aa should be Italic.
Comments on the Quality of English LanguageMinor editing of English language required
Reviewer 2 Report
Comments and Suggestions for Authors
The aim of the evaluated paper is to provide a comprehensive review about the role of infectious and commensal bacteria in the etiopathogenesis of rheumatoid arthritis (RA).
The subject is not new, but it is still one of interest in autoimmune and autoinflammatory diseases.
Introductory part provides some data about rheumatoid arthritis that have to be corrected since they contain some important mistakes:
- “This rheumatic disease manifests itself with many inflammations in joints with different severity
conditions which leads to bone destruction” – what do you mean by “joints with different severity conditions”? and also have in mind that RA is not only about bone destruction (an articular structure is more than that!)
- 2nd paragraph – “One of the major criteria for diagnosis and disease progression is a Disease Activity Score of 28 Joints (DAS28). DAS28 is not only used in RA diagnosis….” This affirmation is quite inaccurate since DAS28 is not used as a major criteria for diagnosis!. Please note that we have nowadays criteria for classification (ACR/EULAR 2010) but not for diagnosis and DAS28 is one of several activity score that can be used.
The discussion about immunopathogenesis is quite well conducted, but nevertheless I don’t understand why the topic about IL17 is significantly larger than about other cytokines more important in RA
- Chapter 5 “Intestinal tract” instead of Track an psoriatic arthritis instead of psoriasis arthritis
- Chapter 6 “Urinary track” please correct spelling
- Please provide explanation for term PBMCs (page 3)
Since the article is described as a comprehensive review please provide in detail the criteria for selecting the articles
Provided conclusions should be more nuanced since till now no major studies have been associated with target of bacterial infections. Also it is mentioned that “Improvements in test results and overall
patient functioning were also obtained with probiotics” but without specific data being discussed in the article.
Reviewer 3 Report
Comments and Suggestions for Authors
Interesting review article on the role of bacterial infections in the pathogenesis of RA.
This is my review:
Introduction
One of the major criteria for diagnosis and disease progression is a Disease Activity Score of 28 Joints (DAS28): DAS 28 is not a criterion for the diagnosis of RA, nor for the progression of the disease. It is an index used to assess RA disease activity. It is necessary to correct the sentence.
Immunopathogenesis of rheumatoid arthritis
Autoimmunological diseases: autoimmune is a better term
Diverse immune cells are implicated in RA pathogenesis (Error! Reference sourcenot found.): I don't understand why this assertion is in the text if it can't be supported with a reference. I think it should be left out
The connection between periodontal disease and rheumatoid arthritis
[28] [26]: correct
Another possibility is that RA predisposes to PD or PD is an RA risk factor due to various pathways, including the activity of periodontal bacteria: This sentence needs to be written more comprehensibly, perhaps separating the statements into separate sentences
Gingipains can be divided into two groups according to the amino acid they are specific to arginine (RgpA or RgpB) and lysine (Kgp): This sentence should be reformulated
In their studies, Svärd et al. discovered that levels of RgpB antibodies in RA patients were significantly higher than in healthy controls. Although they did not find a correlation between these antibodies and ACPA levels of positivity [55]. On the other hand, Bae et al. observed besides the higher levels of RgpB antibodies among RA patients also a positive correlation between them and ACPA le: This part is indistinctly, confusingly written. It should be written more comprehensibly.
Intestinal track microbiota dysbiosis impact on rheumatoid arthritis
...and return of arthritis: unusual formulation, please clarify whether there is a return of arthritis that was in remission...or...
Round 2
Reviewer 1 Report
Comments and Suggestions for Authors
Thank you for responding to my comments and suggestion.
I think that's a big improvement from last one.
There are no more comments and suggestion.
Thanks,
Author Response
Thank you for all your comments and suggestions during this review.
Reviewer 2 Report
Comments and Suggestions for Authors
The authors responded to most of my previous suggestions and now the article is way better, nevertheless I still believe that first two paragraphs need to be rewritten.
"The result is destruction of the bones and cartilages, and the inflammation of synovial membrane by several immune systems" suggestion - the result if the destruction of joint structure
"Progression of the illness can cause systemic disorders in the heart [2], blood vessels [3], kidney [4], liver [5], and lungs" - suggestion - can cause systemic involvement of cardiovascular, respiratory, renal tissue (NB liver involvement is not quite frequent)
Finally I don't understand the role of DAS28 paragraph in the introductory part
